# Metals and Trace Elements in Calcified Valves in Patients with Acquired Severe Aortic Valve Stenosis: Is There a Connection with the Degeneration Process?

**DOI:** 10.3390/jpm13020320

**Published:** 2023-02-13

**Authors:** Aleš Tomášek, Jan Maňoušek, Jan Kuta, Jiří Hlásenský, Leoš Křen, Martin Šindler, Michal Zelený, Petr Kala, Petr Němec

**Affiliations:** 1Centre for Cardiovascular Surgery and Transplantation, Pekařská 53, 656 91 Brno, Czech Republic; 2Department of Internal Cardiology Medicine, University Hospital and Faculty of Medicine, Masaryk University, Jihlavská 20, 625 00 Brno, Czech Republic; 3Research Centre for Toxic Compounds in the Environment (RECETOX), Faculty of Science, Masaryk University, Kamenice 5, 625 00 Brno, Czech Republic; 4Institute of Pathology, University Hospital and Faculty of Medicine, Masaryk University, Jihlavská 20, 625 00 Brno, Czech Republic; 5Institute of Forensic Medicine, St Anne’s University Hospital and Faculty of Medicine, Masaryk University, Tvrdého 2a, 662 99 Brno, Czech Republic

**Keywords:** aortic valve stenosis, calcification, metals and trace elements, biomonitoring, biological effects

## Abstract

Background. Acquired calcified aortic valve stenosis is the most common valve disease in adulthood. In the etiopathogenesis of this complex pathology, the importance of inflammation is mentioned, in which non-infectious influences represented by the biological effects of metal pollutants may participate. The main goal of the study was to determine the concentration of 21 metals and trace elements—aluminium (Al), barium (Ba), cadmium (Cd), calcium (Ca), chrome (Cr), cobalt (Co), copper (Cu), gold (Au), lead (Pb), magnesium (Mg), mercury (Hg), molybdenum (Mo), nickel (Ni), phosphorus (P), selenium (Se), strontium (Sr), sulfur (S), tin (Sn), titanium (Ti), vanadium (V) and zinc (Zn)—in the tissue of calcified aortic valves and to compare them with the concentrations of the same elements in the tissue of healthy aortic valves in the control group. Material and methods. The study group consisted of 49 patients (25 men, mean age: 74) with acquired, severe, calcified aortic valve stenosis with indicated heart surgery. The control group included 34 deceased (20 men, median age: 53) with no evidence of heart disease. Calcified valves were explanted during cardiac surgery and deep frozen. Similarly, the valves of the control group were removed. All valves were lyophilized and analyzed by inductively coupled plasma mass spectrometry. The concentrations of selected elements were compared by means of standard statistical methods. Results. Calcified aortic valves contained significantly higher (*p* < 0.05) concentrations of Ba, Ca, Co, Cr, Mg, P, Pb, Se, Sn, Sr and Zn and—in contrast—lower concentrations of Cd, Cu, Mo, S and V than valves of the control group. Significant positive correlations of concentrations between the pairs Ca-P, Cu-S and Se-S and strong negative correlations between the elements Mg-Se, P-S and Ca-S were found in the affected valves. Conclusion. Aortic valve calcification is associated with increased tissue accumulation of the majority of the analyzed elements, including metal pollutants. Some exposure factors may increase their accumulation in the valve tissue. A relationship between exposure to environmental burden and the aortic valve calcification process cannot be ruled out. Advances in histochemical and imaging techniques allowing imaging of metal pollutants directly in valve tissue may represent an important future perspective.

## 1. Introduction

Acquired calcified aortic valve stenosis is the most common valve disease in adulthood [1]. The exact cause has not been identified yet. Histological studies have shown the development of atherosclerotic-like degenerative changes in the valves of patients with aortic valve stenosis [2,3]. The degeneration process begins with the deposition of lipids in the valve tissue, followed by peroxidation. In the presence of infiltrates of T lymphocytes, monocytes and activated macrophages and with the participation of matrix metalloproteinases, collagen proteolysis, elastin fragmentation and extracellular mass remodeling with signs of inflammation occur [2,3,4,5]. The subsequent activation of fibroblasts and hyalinization, with gradual remodeling of the tissue, can finally lead to the calcification or the ossification of the tissue [4,5,6]. 

Attention is also focused on molecular-biological mechanisms [6,7]. Frequent elevations of C-reactive protein and the presence of some intracellular pathogens (including *Chlamydia pneumoniae*) in these patients suggest the involvement of inflammation or infection [8,9,10]. Other authors question their significance for the degenerative process [11,12]. Inflammatory changes in the tissue can also be caused by the presence of non-infectious agents. These include, among other things, metal pollutants, the particles of which develop biological effects in the tissue associated with alterations in the immune system; oxidative stress; chronic inflammation; and genome changes [13,14,15]. One of the mechanisms of metal-induced inflammation is the delayed type of hypersensitivity reactions demonstrated, for example, in systemic connective tissue diseases [16,17].

The goal of the study was to determine by inductively coupled plasma mass spectrometry the concentration of 21 metals and trace elements—aluminium (Al), barium (Ba), cadmium (Cd), calcium (Ca), chrome (Cr), cobalt (Co), copper (Cu), gold (Au), lead (Pb), magnesium (Mg), mercury (Hg), molybdenum (Mo), nickel (Ni), phosphorus (P), selenium (Se), strontium (Sr), sulfur (S), tin (Sn), titanium (Ti), vanadium (V) and zinc (Zn)—in the tissue of lyophilized calcified aortic valves of patients with severe aortic valve stenosis requiring surgical intervention. These were compared with the concentrations of the same elements in the tissue of lyophilized healthy aortic valves obtained from the autopsy material of deceased patients without heart disease. These major and trace elements were selected for known toxicity, essentiality and abundance in the environment. Therefore, measurable concentrations of the determined metals and trace elements were assumed, even in healthy valves, and higher concentrations of the same elements in calcified valves. The biological effects of metals could be related to the calcification process.

## 2. Material and Methods

The cohort consisted of 49 patients (25 men) of average age 72.44 ± 7.69 years, who were operated on for an acquired, severe, calcified aortic valve stenosis [18]. The echocardiographic parameters of aortic valve stenosis and other characteristics and comorbidities of the patient group are listed in Table 1. The structural changes of the affected valves and the severity of aortic valve stenosis were detected echocardiographically using *Philips Ultrasound (machine) ie33 (Bothell, WA, USA)* [18,19,20]. Left ventricular ejection fraction (LVEF) was calculated by using the Simpson’s biplane method (2D echocardiography), and the transvalvular gradients (maximum and mean) and aortic valve area (AVA) were determined using continuous and pulsed Doppler parameters and instrument software [18,19,20].

Calcified aortic valves were removed during heart surgery using the standard surgical techniques and instruments. The study protocol complied with the Helsinki Declaration and was approved by the Workplace Ethics Committee. All patients completed a *“specially designed Questionnaire on Metal Environmental Burden”* focused on the exposure to common metal pollutants before surgery. In addition to occupational and routine daily exposure, the numbers, types and materials of dental fillings, crowns, bridges, stents, pacemakers or implantable cardioverter-defibrillators, osteosyntheses, endoprostheses and other implants were determined. 

The control group consisted of 34 deceased individuals (20 men) of average age 51.82 ± 17.4 years who underwent pathological-anatomical (4 men and 3 women) or forensic autopsy (16 men and 11 women) and exhibited no macroscopic signs of structural involvement of the heart or aortic valve. A forensic autopsy was performed by law to determine the cause of death in persons who died outside a medical facility in an unexpected or violent manner. It was not possible to obtain all anamnestic, epidemiological or clinical data for these individuals (Table 1).

The following principles were followed in the collection of aortic valves from the patient and control groups: (a) pathological and forensic autopsies were performed no later than 48 h after the death to reduce the effect of autolysis; (b) staff wore talc-free gloves; (c) the edges of the aortic valves were trimmed with ceramic knives after collection; (d) the flaps were placed in polyvinyl chloride (PVC) bags rinsed with demineralized water; (e) tissues were frozen at −80 degrees Celsius; (f) the material was transported for laboratory analysis in thermoboxes containing solid CO_2_; and (g) *St. Thomas* (containing, among others, potassium—K, calcium—Ca, magnesium—Mg and phosphorus—P) cardioplegic solution was used to protect the heart muscle in operated-on patients [21]. To reduce tissue contamination with these ions, the samples of the calcified valves were rinsed with demineralized water (*Aqua Osmotic, Czech Republic, ČSN ISO 3696*) only once. Due to the possible effect of reverse osmosis, the tissues were not soaked in water for a longer period of time [22]. 

The morphology of the aortic valve (number of cusps) was clearly described by the cardiac surgeon in the operating reports, only in 43% of the cases with the predominance of tricuspid valve type. In a total of 49 patients (25 men, 51%), 13 valves (26.5%) were clearly tricuspid (6 men, 12.2%), and 8 valves (16.3%) were bicuspid (4 men, 8.1%). In 28 cases (57.14%), it was not possible to macroscopically determine the type of valve due to the advanced calcification process (15 men, 30.6%). In the control group, a tricuspid aortic valve was present in all subjects (assessed by a pathologist)—Table 1.

After cleaning the valves of residual macroscopic admixture with ceramic knives and lyophilizing the tissue samples (ScanVac CoolSafe, LaboGene, Lillerød, Denmark), the analysis of the elements in the tissue was performed. The concentrations of twenty elements (Al, Au, Ba, Cd, Ca, Co, Cr, Cu, Mg, Mo, Ni, P, Pb, S, Se, Sn, Sr, Ti, V and Zn) were determined after the decomposition of samples with nitric acid and hydrogen peroxide in a microwave digestion system (MWS 3+ Berghof Products + Instruments GmbH, Eningen, Germany), followed by inductively coupled plasma mass spectrometry (Agilent 7700x ICP-MS, Tokio, Japan). Hg content was determined by an AMA254 mercury analyzer directly from solid samples (LECO, St. Joseph, MI, USA). The method was validated by analysis of certified reference materials of biological origin (ERM BB184, ERM BB186 and NIST 1486) and digested aortic valves spiked with a known amount of analytes.

The median recoveries from replicate analysis of the spiked samples and CRMs were found within the range of 93.9 and 110% for spiked samples and 74.9 and 106% for certified reference materials. Typical relative standard deviations within the replicate analysis of spiked samples were found between 1.8 and 10.4%. Data were recorded by single analysis due to the limited amount of samples available for analysis. Limits of detection (LODs) are dependent on the amount of samples available for analysis (usually between 0.2–0.5 g). Typical values of LODs for elements in µg/g are as follows: 0.4 for Al, 0.0005 for Au, 0.01 for Ba, 0.0001 for Cd, 0.0003 for Co, 0.001 for Cr, 0.05 for Cu, 0.005 for Hg, 0.001 for Mo, 0.009 for Ni, 0.002 for Pb, 0.03 for Se, 0.009 for Sn, 0.008 for Sr, 0.06 for Ti, 0.0005 for V and 0.5 for Zn. LODs for elements in mg/g are as follows: 0.0001 for Mg, 0.003 for P, 0.09 for S and 0.002 for Ca.

## 3. Statistical Analysis

Categorical variables were described by absolute and relative frequencies; age was determined by mean and standard deviation. Qualitative data were compared by chi-square and Fisher’s exact test. For continuous variables and due to the highly skewed sample distribution for almost all parameters, robust summary statistics (median, minimum and maximum values) and a nonparametric test (Mann–Whitney U test) were employed for the characterization of metal and trace element levels in tissues and differences in contents between various groups of interest (metal exposition and comorbidities). The Mann–Whitney U test was not used in the case of Al, Au, Hg, Ni, Ti and V, where more than 25% of the results were lower than the method detection limit for a particular element. Spearman’s rank correlation was used to study the associations between particular elements and the age of subjects. 

## 4. Results

The basic characteristics, selected exposure factors and comorbidities in both groups are shown in Table 1. Patients were significantly older than control subjects (*p =* 0.0001), and coronary heart disease in the patient group was present significantly more often (*p =* 0.0001)—Table 1.

Male patients operated on for a severe calcified aortic valve stenosis were significantly younger than women with the same diagnosis (*p =* 0.01). Compared to women, men also had significantly more frequent left ventricular systolic dysfunction (*p =* 0.004) and ischemic heart disease (*p =* 0.042). Other common comorbidities in patients of both sexes were atrial fibrillation, hypertension and dyslipidaemia. Male patients significantly tended to be smokers (*p =* 0.0015) and significantly more often engaged in occupations with a high exposure to metals (*p =* 0.0001) by comparison with women. Heavy metals present a serious health risk to humans when the concentration level and exposure time are increased. According to the questionnaires, the most frequent environmental burden was represented by amalgam dental fillings and dental prostheses, and—to a lesser extent—orthopedic implants. Only a small proportion of patients had implanted coronary stents, pacemakers or implantable cardioverter-defibrillators.

Table 2 shows the results of the analysis of metals and trace elements in the aortic valves of patients and controls. Calcified aortic valves contained significantly higher concentrations (*p* < 0.05) of Ba, Ca, Co, Cr, Pb, Mg, P, Se, Sr, Sn and Zn and, by contrast, less Cd, Cu, Mo and S. For the elements Ba, Ca, Pb, P and Sr, there were tenfold to hundredfold differences in concentrations. For some elements (Al, Au, Hg, Ni, Ti and V), it was only possible to compare the values of the median concentrations for both compared groups, because many samples contained concentrations below the detection limit of the analytical method. The Mann–Whitney U test could not be applied to these datasets. Thus, a twice higher ratio of median concentrations could be observed for gold and twice as low for vanadium. Strong pairwise positive correlations of concentrations (r > 0.6) between the pairs of elements Ca-P, Cu-S and Se-S and strong paired negative correlations of the concentrations Mg-Se, P-S and Ca-S (r > −0.6) between pairs of elements were also found in calcified valves. In non-calcified valves, strong multiple mutual positive correlations (r > 0.6) were found between the elements Ba, Ca, Mg, P, Pb, Sr and Zn; no negative correlations were found here. A positive correlation is understood as a synergy, a negative correlation as an antagonism. Complete correlation matrices for both datasets can be found in the Appendix A. 

The median ratios of element concentrations in the aortic valves of both groups are shown in Table 3. These ratios were significantly higher in patients with a calcified aortic valve and some comorbidities: with dyslipidemia for Zn, with chronic renal insufficiency for Mo and Pb, in women for Cd, Co, Sn and Se and in smokers for Au and Pb. In contrast, the median concentration ratios were significantly lower in patients with a calcified valve and coronary artery disease (CAD) present for Se, malignancies for Cr, smokers for Sn and women for Ca, P, Pb and Zn.

No dependence of elemental concentrations on the age of operated-on patients with calcified valves was found. In contrast, increasing concentrations of Ba, Ca, Co, P, Pb and Sr were identified in uncalcified valves with increasing age (Table 4).

The comparison of the concentrations of elements in the aortic valves of men and women of both groups is presented in Table 5 and Table 6. Although female patients were significantly older than male patients (Table 1), they had significantly lower concentrations of Ca, Mg, P, Pb and Zn in the affected valves and, conversely, significantly higher concentrations of Cd, Co, S, Se and Sn (compared to men).

## 5. Discussion

The concentrations of various metals and trace elements, including metal pollutants, in valve tissue, myocardium, aorta and other tissues have been determined in the past [23,24,25,26]. The association with (heavy) metal exposure has been demonstrated in hypertension, atherosclerosis, endothelial dysfunction and cardiovascular diseases, including dilated cardiomyopathy [14,24,27,28,29]. In general, metals have a high affinity for sulfhydryl groups, inactivating numerous enzymatic reactions, amino acids and sulfur-containing antioxidants, with subsequent decreased oxidant defense and increased oxidative stress. The effects of various metals are described [13,14,15]. For example, Hg and Cd bind to metallothionein and substitute for Cu, Zn and other trace metals, reducing the effectiveness of metalloenzymes. They induce mitochondrial dysfunction, with a reduction in adenosine triphosphate, the depletion of glutathione, an increase of lipid peroxidation and oxidative stress. Other effects include inflammation, thrombosis, vascular smooth muscle dysfunction, endothelial dysfunction, dyslipidemia and immune and mitochondrial dysfunction [13,14,15,28,29]. The detoxification processes of cells (including metals) can be disrupted [30]. These facts are not yet generally accepted in clinical medicine.

An experimental model of calcification is the calciphylaxis described as “a hypersensitivity disorder” [30]. The administration of the so-called “sensitizer” (tissue-sensitizing agent) and “challenger” leads to the induction of calcifications in various organs. “Challengers” include salts of various metals (iron, chromium, aluminium, titanium and lead), as well as corticosteroids and immunosuppressants [30]. A common feature of these substances may be the production of oxidative stress [31]. Another mechanism of metal-induced inflammation represents the delayed type of hypersensitivity reactions demonstrated in systemic connective tissue diseases and in dilated cardiomyopathy [16,17,32].

The calcified aortic valves of our patient group contained significantly higher concentrations (*p* < 0.05) of Ba, Ca, Co, Cr, Mg, P, Pb, Se, Sn, Sr and Zn compared to controls and, conversely, less Cd, Cu, Mo and S. This is comparable to the literature [26]. For some elements (Al, Au, Hg, Ni, Ti and V), it was possible to compare only the values of the median concentrations in both compared groups, because many samples contained concentrations below the detection limit of the analytical method. Differences in concentrations between calcified and healthy valves in the age-unadjusted sample reached up to 120 times the values, and in elderly patients, a maximum of 40 times the values. This indicates increasing concentrations of some elements (Ba, Ca, Co, P, Pb and Sr), even in the tissue of the uncalcified valve with increasing age. However, the concentrations of various elements in the non-calcified valves in controls were lower than those previously described, e.g., in a healthy myocardium [25]. In the aortic valves of healthy individuals, Ca and P concentrations were previously found to be 2.5 times higher than in the mitral valve and up to 19 times higher than in the tricuspid valve. This may indicate an independent accumulation of these elements in the aortic valves [26]. It is not clear why calcification most often affects the aortic valve. The generally accepted explanation may imply more intense mechanical stress on the tissue, with faster blood flow in the high-pressure part of the circulation. Aortic valve stenosis is a highly complex disease, with many contributing factors.

The internal skeleton of the aortic valve is arranged in three layers (fibrosa, spongiosa and ventricularis), which together provide strength to leaflets, resistance to compressive forces and flexibility with changes in the shape of the leaflets during the cardiac cycle. To remain pliable and functional, the aortic valve must undergo continuous repair throughout the whole life [33]. One of the main substrates for collagen biosynthesis is the amino acid proline [34]. The affinity of Cd, Hg and Zn ions to proline was monitored in vitro. In the aqueous phase, this metal ion’s characteristics decreased in the order Zn^2+^ > Cd^2+^ > Hg^2+^ [35]. In another experiment, even 24-h exposure to Rh, Au, Co and Ni salts led to a reduction in collagen synthesis. Salt concentrations found in tissues surrounded by metallic materials were used. The effect of the other ions did not appear within 24 h [36].

Our results also show that with the increasing concentrations of Ca, Mg and P in calcified aortic valves, the concentrations of Se and S decrease. This indicates an increasing predominance of inorganic elements over biogenic ones, especially the loss of the protein component of the tissue during the degeneration process. Elements such as Ba, Mg and Sr can replace Ca in a bond with various forms of calcium phosphate, which is probably due to their similar physicochemical properties (except for P, all others are from group IIA of the periodic table). The higher occurrence of Ca and P (up to tens of percent of dry weight) in calcified tissue is generally due to the presence of amorphous and crystalline forms of calcium phosphate. The association of Mg, Ba and Sr with calcium phosphate has been previously described in urinary calculi due to the cation co-precipitation and cation exchange in the crystal lattice and the sorption to biominerals [37]. Probably not all measured elements enter the structure of mineralized calcium phosphate [38]. This could be indicated by higher concentrations of Al, Cd, Cu, Mo and V in healthy valves, which also contained other pollutants (Table 2).

Lower Cu concentrations in degenerative valves could reflect a decrease in superoxide dismutase activity and an increase in oxidative stress in the tissue [39]. The explanation for the higher concentrations of Zn in calcified valves may be the fact that Zn is part of metalloproteins and enzymes, whose activity increases during inflammation, which is usually found in degenerative valves [2,3,4,8,9]. Inflammation is well-known for decreasing the plasma concentration of Zn due to its higher accumulation in the inflammatory locus and the activation of lymphocytes [40].

An association with heavy metal exposure has been demonstrated in hypertension, atherosclerosis, endothelial dysfunction and cardiovascular disease, for example [27,28]. Pb competes with Ca ions and calcium-binding intracellular proteins and may, thus, adversely affect the activity of Ca-dependent channels and contractile proteins. Pb is also stored in mineralized tissues, with a predilection for bones, in proportion to age, more in men [13,28]. This may explain the higher concentrations of Pb found in calcified valves, where the signs of ossification may be found histologically [5]. According to these facts, the Pb level in calcified valves was increased, especially in smokers and in patients with renal insufficiency [41,42]. 

Significantly lower concentrations of Cd and Mo and, conversely, significantly higher concentrations of Se in the age-unadjusted sample were found in calcified valves compared to controls (Table 2). One possible explanation for these findings could be the different dietary habits of patients and controls; control group individuals were a generation younger. A significant source of Cd are various foods, but also smoking (including passive smoking) [13,41,43]. Cd is associated with renal insufficiency, insulin resistance, mitochondrial dysfunction and cancer [13,43]. As a biogenic element, Mo is contained in certain oxidases. The content of Mo in various foods is determined by its amount in the soil. It is also a component of metal alloys (similar to Cr and W) [44]. In humans, Se is a trace element nutrient that functions as a cofactor for the reduction of antioxidant enzymes, such as glutathione peroxidases [45]. One possible explanation for these findings could be the different dietary habits of patients and controls; control group individuals were a generation younger.

A questionnaire survey revealed significant differences in exposures (duration of years to decades) in the group of patients—men (compared to women) only for smoking and occupational exposure (Table 1). Patients had orthopedic implants, pacemakers, cardioverter-defibrillators or coronary stents for a shorter period of time (months to years); no differences in exposures were found for them.

According to data from questionnaires and autopsy reports, we have not found any association between the presence of orthopedic, dental or other implants containing Ni, Cr and Co and the concentrations of these metals in the calcified valve. Nevertheless, this cannot be ruled out [46,47]. Elevated concentrations of these pollutants have been found in the degenerative valves of younger patients, in whom the presence of metal dental restoration or joint replacements is less common. The corrosion of these materials can lead to the chronic exposure of the released metals [46,47]. In a large group of French patients with total hip arthroplasty, it has recently been shown that individuals with a metal head prosthesis were subsequently more frequently diagnosed with dilated cardiomyopathy or heart failure [48]. Even the presence of pacemakers, implantable cardioverter-defibrillators made of titanium alloy or coronary stents made of alloys of various metals did not correlate with the concentrations of these pollutants in calcified valves [49,50,51].

Individuals from the control group were significantly younger than the patients (Table 1). In an age-adjusted patient sample, significantly increasing concentrations of Ba, Ca, Co, P, Pb and Sr were identified only in health valves with increasing age (Table 4). The higher occurrence of Ba and Sr could be related to the similar physicochemical properties of these elements (group IIA of the periodic table including Ca), which may be represented or accompanied in chemical compounds. Pb can also compete with Ca ions [37,38], and Co can adsorb onto mineralized forms of calcium phosphate. Yet, calcifications did not occur in healthy valves. Any incipient calcifications or microcalcifications could have been demonstrated histopathologically, microscopically and electron microscopically [52]. We did not use any such techniques.

Although our female patients with calcified aortic valves were significantly older than the male patients (Table 1), they had significantly lower concentrations of Ca, Mg, P, Pb and Zn in the affected valves and, conversely, significantly higher concentrations of Cd, Co, S, Se and Sn (compared to men). This may indicate less advanced valvular calcification in operated-on women, with a higher proportion of the original organic component of the valve. Elevated Sn concentrations in patients’ valves can be explained by exposure to dental amalgam fillings, which are the most common source of Sn in the Czech population. According to the questionnaires, these were found in 84% of men and 92% of women in the patient group. Another source of Sn may be occupational exposure, which was significantly more common in men (Table 1).

Today, alloys of various metals are widely used in products of daily use, but also in implantology and prosthetics [53,54,55,56]. Despite the effort for maximum biocompatibility, these materials are not inert and are subject to gradual corrosion [46,47,48,53,54,55]. Metal pollutants enter the body also through breathing and food [41,56]. In the tissues, they develop biological effects (including delayed-type hypersensitivity reactions), and may pose a health risk [13,14,47,57]. The effort should be the development and use of more compatible and more inert materials in various areas of life, including medicine, with the aim of limiting the mentioned health risks. However, they will continue to be present in the natural environment.

## 6. Conclusions

Our paper presents data on the concentrations of metal and trace elements in the tissue of calcified and non-degenerate aortic valves of the Czech population. The most common comorbidities of our patients with severe calcified aortic valve stenosis requiring surgical treatment are ischemic heart disease, left ventricular systolic dysfunction, atrial fibrillation, hypertension and dyslipidemia. Amalgam dental fillings represent the most common environmental burden in patients and controls; no significant differences in exposure to metals were found. 

Aortic valve calcification is associated with increased tissue accumulation of most of the elements determined (Mg, P, Ca, Cr, Co, Zn, Se, Sr, Ba, Pb and Sn). For the elements P, Ca, Sr, Ba and Pb, there are tenfold to hundredfold differences in concentrations compared to the control group. The correlation between the elements in the calcified valve tissue also shows that with increasing Ca, P and Mg content, the S and Se concentrations decrease. This indicates the predominance of inorganic elements over biogenic ones, especially the loss of the protein component of the valve tissue during calcification. In an age-adjusted patient sample, significantly increasing concentrations of Ba, Ca, Co, P, Pb and Sr were identified only in health valves with increasing age. Compared to men, female patients with calcified aortic valves had significantly lower concentrations of Ca, Mg, P, Pb and Zn in the affected valves and, conversely, significantly higher concentrations of Cd, Co, Cd, Co, S, Se and Sn. This may indicate less advanced valve calcification in operated-on women, with a higher proportion of the original organic component of the valve. A relationship between exposure to environmental burden and the aortic valve calcification process cannot be ruled out.

The homogenization of valve samples in our work did not allow us to determine the exact localization of metal ions or their binding sites in the valve tissue. Advances in histochemical and imaging techniques, as well as assessment of the intensity of oxidative stress associated with the biological effects of metals, could help to better understand the degeneration process, including inter-individual differences [56,57]. Apparently, the influence of gender also plays a role.

## 7. Limitation

We acknowledge that this is a single-center study with a relatively small number of patients and controls. Individuals from the control group were significantly younger than the patients, and the results of the analyses were adjusted for age.

It was not possible to obtain all anamnestic, epidemiological or clinical data for individuals in the control group (Table 1). However, neither the aortic valves nor the heart tissues of the control subjects showed macroscopic signs of disease. 

Ca, Mg and P concentrations in calcified valves could be overestimated using cardioplegia. The concentrations of other elements (Al, Hg, Ni and Ti) in a large part of the samples were below the detection limit of the analytical method. However, this does not mean that they cannot develop a specific biological effect in the tissue. The results of the analysis can be influenced by a number of factors, which we tried to limit when taking the valves.

## Figures and Tables

**Table 1 jpm-13-00320-t001:** The basic characteristics, comorbidities and selected exposure factors in the groups of patients and controls (including the comparison of men and women in both groups).

Characteristics, Comorbidity, Exposure Factor	Patients (*n* = 49)		*p* *	Controls (*n* = 34)		*p* *	*p* **
	Men (*n* = 25)	Women (*n* = 24)	0.999	Men (*n* = 20)	Women (*n* = 14)	0.225	0.5101
Age (years)	70.8 ± 10.1	73.4 ± 5.9	0.01	50.25 ± 18.88	54.07 ± 15.45	0.5981	0.0001
BMI (kg/m^2^)	30.0 ± 5.4	30.2 ± 5.4	0.999	n/a	n/a	n/a	n/a
Morphology of the aortic valve—number of cusps							
-Tricuspid	6 (12.2%)	7 (14.3%)	0.753	20 (58.8%)	14 (41.2%)	0.999	0.0001
-Bicuspid	4 (8.1%)	4 (8.1%)	0.999	0 (0%)	0 (0%)	0.999	0.0186
-Cannot be determined	15 (30.6%)	13 (26.5%)	0.776	0 (0%)	0 (0%)	0.999	0.0001
Aortic valve area—AVA (cm^2^)	0.8 ± 0.2	0.8 ± 0.2	0.999	n/a	n/a	n/a	n/a
Aortic valve area index—AVAi (cm^2^/m^2^)	0.4 ± 0.1	0.4 ± 0.1	0.999	n/a	n/a	n/a	n/a
Aortic valve gradient maximal (mmHg)	72.9 ± 16.5	73.4 ± 15.5	0.98	n/a	n/a	n/a	n/a
Aortic valve gradient mean (mmHg)	44.2 ± 11.4	44.9 ± 10.6	0.98	n/a	n/a	n/a	n/a
Left ventricular ejection fraction—LVEF (%)	55.6 ± 12.4	61.4 ± 7.6	0.07	n/a	n/a	n/a	n/a
Left ventricular ejection fraction—LVEF < 50%	8 (16.3%)	0 (0%)	0.004	n/a	n/a	n/a	n/a
Comorbidity							
-Thoracic aorta dilatation ≥ 41 mm	5 (10.2%)	5 (10.2%)	0.999	n/a	n/a	n/a	n/a
-Coronary artery disease (CAD)	19 (38.7%)	11 (22.4%)	0.042	0 (0%)	0 (0%)	0.999	0.0001
-Atrial fibrillation (all forms)	23 (46.9%)	21 (42.8%)	0.667	n/a	n/a	n/a	n/a
-Hypertension	21 (42.8%)	23 (46.9%)	0.609	n/a	n/a	n/a	n/a
-Dyslipidemia	17 (34.6%)	16 (32.6%)	0.999	n/a	n/a	n/a	n/a
-Diabetes mellitus	12 (24.4%)	9 (18.3%)	0.567	n/a	n/a	n/a	n/a
-Malignancy	2 (4%)	3 (6.1%)	0.67	0 (0%)	0 (0%)	0.999	0.0753
-Allergy	6 (12.2%)	8 (16.3%)	0.47	n/a	n/a	n/a	n/a
Exposure factor							
-Amalgam filling	21 (42.8%)	22 (44.9%)	0.667	17 (50%)	12 (35.29%)	0.999	0.7532
-False teeth (plastic with Ni-Co-Cr wire)	13 (26.5%)	14 (28.5%)	0.776	7 (20.6%)	5 (14.7%)	0.999	0.1170
-Orthopedic implant	4 (8.1%)	5 (10.2%)	0.725	1 (2.9%)	1 (2.9%)	0.999	0.1861
-Pacemaker, implantable cardioverter-defibrillator	2 (4%)	1 (2%)	0.999	0 (0%)	0 (0%)	0.999	0.2656
-Coronary stent	5 (10.2%)	1 (2%)	0.189	0 (0%)	0 (0%)	0.999	0.0767
-Occupational exposure ^1^	18 (36.7%)	4 (8.1%)	0.0001	n/a	n/a	n/a	n/a
-Smoking	13 (26.5%)	2 (4%)	0.0015	n/a	n/a	n/a	n/a

(%)—Relative numbers of the total number of 49 patients and 34 controls, respectively. *p* *—comparison of male and female patients, male and female controls, respectively. *p* **—comparison of patients and controls. ^1^ Occupational exposure—foundry, metalworking, welding, soldering and metalsmithing.

**Table 2 jpm-13-00320-t002:** Concentrations of metals and trace elements in the tissue of aortic valves of patients and controls.

Element	Unit	Patients (*n* = 49)		Controls (*n* = 34)		Median Ratio (P/C)	*p*
		Median	Min.—Max.	Median	Min.—Max.		
Al	µg/g	0.4	<0.4–2.3	0.6	<0.4–20.0	0.7	n/a
Au	µg/g	0.0010	<0.005–0.0349	<0.0005	<0.0005–0.0552	>2	n/a
Ba	µg/g	4.57	2.08–11.6	0.112	0.03–1.53	40.8	<0.0001
Ca	mg/g	177 *	78.9–243 *	1.53	0.38–50.9	115.7 *	<0.0001 *
Cd	µg/g	0.0266	0.00355–0.115	0.1350	0.0191–0.583	0.2	<0.0001
Co	µg/g	0.0255	0.0063–0.112	0.0157	0.0044–0.149	1.6	0.0375
Cr	µg/g	0.055	0.010–0.264	0.025	0.008–0.23	2.2	0.0014
Cu	µg/g	0.79	0.40–2.42	2.09	0.55–4.44	0.4	<0.0001
Hg	µg/g	<0.005	<0.005–0.018	<0.005	<0.005–0.025	n/a	n/a
Mg	mg/g	2.24 *	1.01–3.03 *	0.247	0.0914–0.994	9.1 *	<0.0001 *
Mo	µg/g	0.0086	0.0043–0.0254	0.0170	0.0065–0.0674	0.5	<0.0001
Ni	µg/g	0.012	<0.009–0.110	0.011	<0.009–0.11	1.1	n/a
P	mg/g	86.9 *	38.6–119 *	2.37	1.36–29.4	36.7 *	<0.0001 *
Pb	µg/g	0.863	0.285–4.24	0.069	0.005–8.77	12.6	<0.0001
S	mg/g	3.30	2.23–5.29	3.89	1.15–5.27	0.8	0.0199
Se	µg/g	0.345	0.195–0.829	0.296	0.118–0.436	1.2	0.0067
Sn	µg/g	0.227	0.032–2.63	0.116	0.011–1.08	2.0	0.0008
Sr	µg/g	59.5	29.7–125	0.969	0.207–27.90	61.4	<0.0001
Ti	µg/g	<0.06	<0.02–0.34	<0.06	<0.03–0.55	n/a	n/a
V	µg/g	0.0010	<0.0005–0.0032	0.0022	<0.0005–0.0168	0.5	n/a
Zn	µg/g	117	55–180	30.6	12.1–51.5	3.8	<0.0001

* Disputable effect of cardioplegia (*St. Thomas* solution); μg/g (microgram per gram), mg/g (milligram per gram). Calcified valves contained significantly higher concentrations (*p* < 0.05) of Ba, Ca, Co, Cr, Pb, Mg, P, Se, Sr, Sn and Zn and, by contrast, less Cd, Cu, Mo and S. For Ba, Ca, Pb, P and Sr, there were tenfold to hundredfold differences in concentrations. The concentrations of Al, Au, Hg, Ni, Ti and V in the samples were below the detection limit of the analytical method.

**Table 3 jpm-13-00320-t003:** Median concentration ratios of metals and trace elements in aortic valves of patients and controls in selected comorbidities and characteristics.

Comorbidity, Characteristics (Patient Group)	Element	Median Concentration Ratio (>1)	*p*	Element	Median Concentration Ratio (<1)	*p*
		Patients/Controls			Patients/Controls	
Coronary artery disease (CAD)				Se	0.3185/0.389	0.001
Dyslipidaemia	Zn	125/103	0.0063			
Chronic kidney disease (CKD)	Mo	0.01035/0.008	0.0104			
	Pb	1.16/0.7885	0.002			
Malignancy				Cr	0.013/0.06	0.057
Smoking	Au	0.0015/0.0009	<0.0001	Sn	0.104/0.335	0.0002
	Pb	1.19/0.796	<0.0001			
Gender—women	Cd	0.0357/0.0233	0.0299	Ca	167.5/201 *	0.0005
	Co	0.0318/0.019	<0.0001	P	82.5/97.6 *	0.0009
	Se	0.385/0.293	0.0002	Pb	0.796/1.19	0.013
	Sn	0.433/0.112	0.0062	Zn	106.5/131	0.013
Gold crowns				Au	0.0005/0.0014	<0.0001

* unit—mg/g; (milligram per gram); all other concentrations in mg/g (microgram per gram). Median concentration ratios were significantly higher in patients with calcified valve and: with dyslipidemia for Zn, with CKD for Mo and Pb, in women for Cd, Co, Sn and Se and in smokers for Au and Pb. By contrast, they were significantly lower in patients with calcified valve and CAD present for Se, malignancies for Cr, in smokers for Sn and in women for Ca, P, Pb and Zn.

**Table 4 jpm-13-00320-t004:** Spearman’s rank correlation between age of donors and content of metals and trace elements in aortic valve samples.

	Patients		Controls	
Element	Spearman r	*p* (Two-Tailed)	Spearman r	*p* (Two-Tailed)
Ba	−0.087	0.548	0.571	<0.001
Ca	0.036	0.803	0.651	<0.001
Cd	0.031	0.830	0.321	0.064
Co	0.130	0.372	0.439	0.009
Cr	−0.176	0.226	0.279	0.109
Cu	−0.166	0.253	−0.230	0.189
Mg	0.028	0.848	0.319	0.065
Mo	0.186	0.199	0.0224	0.899
P	0.008	0.954	0.462	0.005
Pb	0.047	0.744	0.558	0.001
S	−0.062	0.670	−0.241	0.169
Se	−0.047	0.743	−0.179	0.311
Sn	−0.055	0.704	0.0401	0.821
Sr	0.114	0.431	0.771	<0.001
Zn	0.117	0.420	0.098	0.580

Increasing concentrations of Ba, Ca, Co, P, Pb and Sr were identified in controls with healthy valves with increasing age.

**Table 5 jpm-13-00320-t005:** Concentrations of metals and trace elements in the tissue of calcified aortic valves of patient group—comparison of men and women.

Element	Unit	Men (*n* = 25)		Women (*n* = 24)		Median Ratio (M/W)	*p*
		Median	Min.—Max.	Median	Min.—Max.		
Al	µg/g	0.6	<0.4–1.4	0.7	<0.4–2.3	0.9	n/a
Au	µg/g	0.0014	<0.005–0.0315	0.00145	<0.005–0.0349	1.0	n/a
Ba	µg/g	4.29	2.56–9.45	4.645	2.08–11.6	0.9	0.7851
Ca	mg/g	189.0 *	78.9–243 *	169.0 *	99.8–211 *	1.1 *	0.0046 *
Cd	µg/g	0.0208	0.00355–0.079	0.03475	0.016–0.115	0.6	0.0103
Co	µg/g	0.0168	0.0063–0.0701	0.03155	0.0103–0.112	0.5	<0.0001
Cr	µg/g	0.05	0.010–0.264	0.058	0.011–0.201	0.9	0.059
Cu	µg/g	0.75	0.40–2.42	0.85	0.58–1.42	0.9	0.0512
Hg	µg/g	0.006	<0.005–0.018	0.001	<0.005–0.002	n/a	n/a
Mg	mg/g	2.39 *	1.01–3.03 *	2.195 *	1.14–2.96 *	1.1 *	0.0242 *
Mo	µg/g	0.0087	0.0043–0.0254	0.0082	0.0055–0.0132	1.1	0.5223
Ni	µg/g	0.02	<0.009–0.03	0.018	0.01–0.11	1.1	n/a
P	mg/g	91.9 *	38.6–119 *	83.45 *	49.1–106.0 *	1.1 *	0.0059 *
Pb	µg/g	1.09	0.5–4.24	0.7885	0.285–3.8	1.4	0.0327
S	mg/g	3.13	2.23–4.97	3.49	2.8–5.29	0.9	0.0016
Se	µg/g	0.317	0.195–0.829	0.3850	0.3–0.482	0.8	0.0033
Sn	µg/g	0.184	0.032–0.977	0.383	0.06–2.63	0.5	0.0157
Sr	µg/g	58.3	29.7–125	60.55	36.2–118	1.0	0.9487
Ti	µg/g	0.07	0.03–0.27	0.07	0.02–0.34	1.0	0.3843
V	µg/g	0.0014	<0.0005–0.0029	0.0012	<0.0005–0.0032	1.2	n/a
Zn	µg/g	125.0	55–180	107.5	73.0–173	1.2	0.0191

* Disputable effect of cardioplegia (*St. Thomas* solution); μg/g (microgram per gram), mg/g (milligram per gram). Although female patients were significantly older than male patients, they had significantly lower concentrations of Ca, Mg, P, Pb and Zn in the affected valves and, conversely, significantly higher concentrations of Cd, Co, S, Se and Sn (compared to men).

**Table 6 jpm-13-00320-t006:** Concentrations of metals and trace elements in the tissue of healthy aortic valves—comparison of men and women.

Element	Unit	Men (*n* = 20)		Women (*n* = 14)		Median Ratio (M/W)	*p*
		Median	Min.—Max.	Median	Min.—Max.		
Al	µg/g	0.75	0.3–20	0.95	0.2–2.2	0.8	0.3401
Au	µg/g	<0.005	<0.005–0.055	<0.005	<0.005	n/a	n/a
Ba	µg/g	0.1325	0.032–1.53	0.0905	0.03–0.34	1.5	0.0618
Ca	mg/g	1.95	0.44–50.9	1.525	0.38–4.74	1.3	0.4311
Cd	µg/g	0.132	0.0191–0.55	0.1955	0.0496–0.583	0.7	0.1101
Co	µg/g	0.01425	0.0044–0.149	0.02025	0.0086–0.0431	0.7	0.4109
Cr	µg/g	0.023	0.009–0.23	0.032	0.008–0.066	0.7	0.528
Cu	µg/g	1.99	1.22–3.03	2.15	0.549–4.44	0.9	0.528
Hg	µg/g	0.0065	<0.005–0.016	0.011	<0.005–0.025	n/a	n/a
Mg	mg/g	0.255	0.149–0.994	0.239	0.0914–0.379	1.1	0.3274
Mo	µg/g	0.0178	0.0097–0.0374	0.01635	0.0065–0.0674	1.1	0.6346
Ni	µg/g	0.012	<0.009–0.11	0.022	<0.009–0.057	n/a	n/a
P	mg/g	2.385	1.36–29.4	2.255	1.57–3.93	1.1	0.421
Pb	µg/g	0.1085	0.012–2.57	0.054	0.005–8.77	2.0	0.3632
S	mg/g	3.82	2.47–5.27	3.93	1.15–4.93	1.0	0.8427
Se	µg/g	0.2885	0.187–0.436	0.3255	0.118–0.429	0.9	0.6472
Sn	µg/g	0.1105	0.011–0.558	0.131	0.022–1.08	0.8	0.9656
Sr	µg/g	0.974	0.392–27.9	0.939	0.207–2.66	1.0	0.3275
Ti	µg/g	0.08	0.04–0.55	0.06	0.03–0.16	1.3	0.99
V	µg/g	0.0022	0.001–0.0168	0.0029	<0.0005–0.0155	n/a	n/a
Zn	µg/g	31.15	17.8–51.5	30.15	12.1–43.1	1.0	0.6103

μg/g (microgram per gram), mg/g (milligram per gram).

## Data Availability

The data presented in this study are available on request from the corresponding author. The data are not publicly available due to the policy of Centre for Cardiovascular Surgery and Transplantation, Brno, Czech Republic.

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
