# Peer review of "Metals and Trace Elements in Calcified Valves in Patients with Acquired Severe Aortic Valve Stenosis: Is There a Connection with the Degeneration Process?"

_jpm, 2023, doi:10.3390/jpm13020320_

Round 1

Reviewer 1 Report

Manuscript title:  Metals and trace elements in calcified valves in patients with  acquired severe aortic valve stenosis: is there a connection with  the degeneration process?   

Manuscript id: jpm-2109783

Authors:   Tomášek  et al.

The manuscript is particularly strong regarding the less studied topic and the experimental setup on heavy elements among patients…. The manuscript regarding the topic and results presented is of interest to the public health community and revisions based on the comments below are recommended before considering for publication.

Major comments

·       Insufficient Abstract: In the abstract, the main aim and background of the manuscript are missing, the current version it only highlights the result. In addition, it would be even better to have a sentence as a future perspective.

·       The unit/abbreviation is not mentioned before, consider defining the abbreviation when mentioned for the first time…. Please check throughout the manuscript to define the abbreviations.

·       Line 98-107, the aim or hypothesis of the study is clear, however, the approach is missing ….

·       Lake of scientific literature to support the statements and findings throughout the manuscript…... I have made some suggestions for that and more need it….

·       More information is needed for ALL TABLE captions and define the abbreviation and units that are used. And adjust the significant figures for the table and manuscript.

·       I am not sure whether the ‘’…..’’ term is well discussed in the abstract and manuscript. Please consider discussing it or rephrasing it.

·       I have a major concern about the results and discussion section. The authors describe the results and compare the results with previous studies, however, insight mechanisms are still insufficient.

·       The discussion regarding haravy an dtrace element level with the diseases is missing? Or why simply why we should monitor metal and dtrace elements among the patients

Detailed comments

abstract

If the unit/abbreviation is not mentioned before, consider defining the abbreviation wtracentioned for the first time.

Introduction:

Line 81-87: A complicated sentence, please revise and check the grammar

Line 92: A reference is needed here.

In MM section

Additional info is needed for the table caption, most importantly significant figures.

In MM section, what is the quality control (QC) data? There is no mention of the QC.

What is the accuracy of the instruments, recovery, LOD, and LOQ ……. These parameters are needed to report the efficiency of any analytical system.

In general, how many times you’ve recorded the data,? duplicate? Triplicate?..... what you mentioned in the text is not clear, please elaborate more on this

R&D section

Additional info is needed for the table caption, most importantly significant figures.

Line 245: consider using this reference: https://doi.org/10.1007/s10661-020-08818-w

These sections are repeating information already presented and explain things in an unnecessarily complicated way. The quality of the manuscript would benefit from the whole section being condensed, Line 259-286 and Line 327-375.

Conclusion

Important conclusions! However, the future perspectives for the following research are highly crucial here …..

Author Response

Reviewer 1

The manuscript is particularly strong regarding the less studied topic and the experimental setup on heavy elements among patients. The manuscript regarding the topic and results presented is of interest to the public health community and revisions based on the comments below are recommended before considering for publication.

Major comments

  • Insufficient Abstract: In the abstract, the main aim and background of the manuscript are missing, the current version it only highlights the result. In addition, it would be even better to have a sentence as a future perspective.
  • The unit/abbreviation is not mentioned before, consider defining the abbreviation when mentioned for the first time…. Please check throughout the manuscript to define the abbreviations.
  • Line 98-107, the aim or hypothesis of the study is clear, however, the approach is missing ….

Response:The text of both the Abstract and the Introduction have been edited with visible changes.

  • Lake of scientific literature to support the statements and findings throughout the manuscript…... I have made some suggestions for that and more need it….

Response: Answered below in detailed comments.

More information is needed for ALL TABLE captions and define the abbreviation and units that are used. And adjust the significant figures for the table and manuscript.

Response: Abbreviations and units request resolved. Important results (numbers) from the tables are inserted into the text of the article. Additional information has been added to Tables 2-6.

I am not sure whether the ‘’…..’’ term is well discussed in the abstract and manuscript. Please consider discussing it or rephrasing it. · I have a major concern about the results and discussion section. The authors describe the results and compare the results with previous studies, however, insight mechanisms are still insufficient.

Response: Answered below in detailed comments with visible changes in the text of the manuscript.

The discussion regarding heavy and trace element level with the diseases is missing? Or why simply why we should monitor metal and dtrace elements among the patients.

Response: An association with (heavy) metal exposure has been demonstrated in hypertension, atherosclerosis, endothelial dysfunction and cardiovascular diseases, including dilated cardiomyopathy (14,23,26–28). In general, metals have a high affinity for sulfhydryl groups, inactivating numerous enzymatic reactions, amino acids, and sulfur-containing antioxidants, with subsequent decreased oxidant defense and increased oxidative stress. The effects of various metals are described (13–15). E.g. Hg and Cd bind to metallothionein and substitute for Cu, Zn and other trace metals reducing the effectiveness of metalloenzymes. They induce mitochondrial dysfunction with the reduction in adenosine triphosphate, the depletion of glutathione, increase lipid peroxidation and oxidative stress. Other effects include inflammation, thrombosis, vascular smooth muscle dysfunction, endothelial dysfunction, dyslipidemia, immune and mitochondrial dysfunction (13–15,27,28). The detoxification processes of cells (including metals) can be disrupted (29). These facts are not yet generally accepted in clinical medicine.

References

  1. Bondy SC. Metal Toxicity, Inflammation and Oxidative Stress. In: Bondy SC, Campbell A, editors. Inflammation, Aging, and Oxidative Stress [Internet]. Cham: Springer International Publishing; 2016 [cited 2021 Jul 25]. p. 3–16. (Oxidative Stress in Applied Basic Research and Clinical Practice). Available from: http://link.springer.com/10.1007/978-3-319-33486-8_1
  2. Flora SJS. Toxic metals: Health effects, and therapeutic measures. J Biomed Ther Sci. 2014(1):48–64.
  3. Ray PD, Yosim A, Fry RC. Incorporating epigenetic data into the risk assessment process for the toxic metals arsenic, cadmium, chromium, lead, and mercury: strategies and challenges. Front Genet [Internet]. 2014 Jul 16 [cited 2021 Jul 25];5. Available from: http://journal.frontiersin.org/article/10.3389/fgene.2014.00201/abstract
  4. Frustaci A, Magnavita N, Chimenti C, Caldarulo M, Sabbioni E, Pietra R, et al. Marked elevation of myocardial trace elements in idiopathic dilated cardiomyopathy compared with secondary cardiac dysfunction. J Am Coll Cardiol. 1999 May;33(6):1578–83.
  5. Saleh MQ, Hamad ZA, Hama JR. Assessment of some heavy metals in crude oil workers from Kurdistan Region, northern Iraq. Environ Monit Assess. 2021 Jan;193(1):49.
  6. Navas-Acien A, Guallar E, Silbergeld EK, Rothenberg SJ. Lead Exposure and Cardiovascular Disease—A Systematic Review. Environ Health Perspect. 2007 Mar;115(3):472–82.
  7. Houston MC. The role of mercury and cadmium heavy metals in vascular disease, hypertension, coronary heart disease, and myocardial infarction. Altern Ther Health Med. 2007 Apr;13(2):S128-133.
  8. Lushchak VI. Glutathione Homeostasis and Functions: Potential Targets for Medical Interventions. J Amino Acids. 2012 Feb 28;2012:1–26.

Detailed comments:

Abstract

If the unit/abbreviation is not mentioned before, consider defining the abbreviation wtracentioned for the first time.

Response: In the etiopathogenesis of this complex pathology, the importance of inflammation is mentioned, in which non-infectious influences represented by the biological effects of metal pollutants may participate. The main goalof the study was to determine the concentration of 21 metals and trace elements – aluminium (Al), barium (Ba), cadmium (Cd), calcium (Ca), chrome (Cr), cobalt (Co), copper (Cu), gold (Au), lead (Pb), magnesium (Mg), mercury (Hg), molybdenum (Mo), nickel (Ni), phosphorus (P), selenium (Se), strontium (Sr), sulphur (S), tin (Sn), titanium (Ti), vanadium (V) and zinc (Zn) – in the tissue of calcified aortic valves and to compare them with the concentrations of the same elements in the tissue of healthy aortic valves in the control group.

Advance in histochemical and imaging techniques allowing imaging of metal pollutants directly in valve tissue may represent an important future perspective.

Corrected in the Abstract text with visible changes.

Introduction

Line 81-87: A complicated sentence, please revise and check the grammar.

Response: The degeneration process begins with the deposition of lipids in the valve tissue, followed by peroxidation. In the presence of infiltrates of T lymphocytes, monocytes and activated macrophages and with the participation of matrix metalloproteinases, collagen proteolysis, elastin fragmentation and extracellular mass remodeling with the signs of inflammation occur (2–5). The subsequent activation of fibroblasts and hyalinization with the gradual remodeling of the tissue can finally lead to the calcification or the ossification of the tissue (4-6).

Reworded as requested.

References:

  1. Lis GJ, Czapla-Masztafiak J, Kwiatek WM, Gajda M, Jasek E, Jasinska M, et al. Distribution of selected elements in calcific human aortic valves studied by microscopy combined with SR-μXRF: Influence of lipids on progression of calcification. Micron. 2014 Dec;67:141–8.
  2. 0tto CM, Kuusisto J, Reichenbach DD, Gown AM, O’Brien KD. Characterization of the early lesion of ‘degenerative’ valvular aortic stenosis. Histological and immunohistochemical studies. Circulation. 1994 Aug;90(2):844–53.
  3. Rajamannan NM, Subramaniam M, Rickard D, Stock SR, Donovan J, Springett M, et al. Human Aortic Valve Calcification Is Associated With an Osteoblast Phenotype. Circulation. 2003 May 6;107(17):2181–4.
  4. Fondard O, Detaint D, Iung B, Choqueux C, Adle-Biassette H, Jarraya M, et al. Extracellular matrix remodelling in human aortic valve disease: the role of matrix metalloproteinases and their tissue inhibitors. Eur Heart J. 2005 Jul 1;26(13):1333–41.
  5. Wirrig EE, Hinton RB, Yutzey KE. Differential expression of cartilage and bone-related proteins in pediatric and adult diseased aortic valves. J Mol Cell Cardiol. 2011 Mar;50(3):561–9.

Line 92: A reference is needed here.

Response: Inflammatory changes in the tissue  can also be caused by the presence of non-infectious agents (13).

Reference

  1. Bondy SC. Metal Toxicity, Inflammation and Oxidative Stress. In: Bondy SC, Campbell A, editors. Inflammation, Aging, and Oxidative Stress [Internet]. Cham: Springer International Publishing; 2016 [cited 2021 Jul 25]. p. 3–16. (Oxidative Stress in Applied Basic Research and Clinical Practice). Available from: http://link.springer.com/10.1007/978-3-319-33486-8_1

In MM section

Additional info is needed for the table caption, most importantly significant figures.

In MM section, what is the quality control (QC) data? There is no mention of the QC.

What is the accuracy of the instruments, recovery, LOD, and LOQ ……. These parameters are needed to report the efficiency of any analytical system.

In general, how many times you’ve recorded the data,? duplicate? Triplicate?..... what you mentioned in the text is not clear, please elaborate more on this

Response: Following text was added to the manuscript (line 164):

Median recoveries from replicate analysis of spiked samples and CRMs were found within the range of 93.9 and 110 % for spiked samples and 74.9 and 106 % for certified reference materials. Typical relative standard deviations within the replicate analysis of spiked samples were found between 1.8 and 10.4 %. Data were recorded by single analysis due to limited amount of sample available for analysis. Limits of detection (LODs) are dependent on amount of sample available for analysis (usually between 0.2 – 0.5 g). Typical values of LODs for elements in µg/g are as follows: 0.4 for Al, 0.0005 for Au, 0.01 for Ba, 0.0001 for Cd, 0.0003 for Co, 0.001 for Cr, 0.05 for Cu, 0.005 for Hg, 0.001 for Mo, 0.009 for Ni, 0.002 for Pb, 0.03 for Se, 0.009 for Sn, 0.008 for Sr, 0.06 for Ti, 0.0005 for V and 0.5 for Zn. LODs for elements in mg/g are as follows: 0.0001 for Mg, 0.003 for P, 0.09 for S and 0.002 for Ca.

R&D section

Additional info is needed for the table caption, most importantly significant figures. Line 245: consider using this reference: https://doi.org/10.1007/s10661-020-08818-w

These sections are repeating information already presented and explain things in an unnecessarily complicated way. The quality of the manuscript would benefit from the whole section being condensed, Line 259-286 and Line 327-375.

Response: Additional informations has been added to Tables 2-6 as  requested.

In line 245, the required reference was added.

The text in the range of lines 259-286 and lines 327-375 has been shortened as requested.

Conclusion

Important conclusions! However, the future perspectives for the following research are highly crucial here …..

Response: The text of the conclusion has been expanded as requested.

The homogenization of valve samples in our work did not allow us to determine the exact localization of metal ions or their binding sites in the valve tissue. Advances in histochemical and imaging techniques as well as the assessment of the intensity of oxidative stress associated with the biological effects of metals could help to better understand the degeneration process including inter-individual differences (58,59). Apparently, the influence of gender also plays a role.

References

  1. McRae R, Bagchi P, Sumalekshmy S, Fahrni CJ. In Situ Imaging of Metals in Cells and Tissues. Chem Rev. 2009 Oct 14;109(10):4780–827.
  2. Greenberg HZE, Zhao G, Shah AM, Zhang M. Role of oxidative stress in calcific aortic valve disease and its therapeutic implications. Cardiovasc Res. 2022 May 6;118(6):1433–51.

Reviewer 2 Report

Tomášek and colleagues present a simple, but well-designed, study characterising the content of calcified aortic valves in metals and trace elements using ICP-MS. The authors provide a detailed characterisation of the patients studied. The cohort is sex-balanced, and this is key, considering the major sex differences in aortic stenosis presentation (women tend to present higher levels of fibrosis, and men tend to present higher levels of calcification for the same level of stenosis). The age effect, a major confounder and source of bias, was also appraised by means of correlations. I also found the definition and listing of other potential confounding factors pertinent, including occupational exposure or the presence of potential sources of exogenous metals/trace elements, such as in amalgam fillings. The sample collection and treatment protocol were also adequate for studying the metals/trace elements, minimising the risk of contamination.

Although no major translational outputs have been taken from this study, I believe this paper is an important contributor to the field in the sense that it provides important data that helps elucidate the role of other elements apart from calcium and phosphorous in the process of aortic valve degeneration. It can be helpful to other researchers exploring sex- and age-specific diagnostic and therapeutic approaches for aortic valve stenosis.

I recommend the publication of the manuscript, should the authors address only these minor points:

The study design and rationale are well explained, but I missed some details. Particularly:

1.       The variable ‘Occupational exposure’ should be better explained: what kind of jobs/activities were considered as an exposure risk? Add this to the manuscript.

2.       The echocardiographic methods for assessing aortic stenosis severity and ejection fraction calculation should be indicated (for instance, did you use 2D or 3D echo? Did you use tissue Doppler? Did you use the modified Bernoulli equation to obtain the transvalvular gradients? Did you use Simpson’s biplane method for EF quantification? Did you follow the EAE guidelines?).

3.       The ICP-MS method used should be described in more detail (or at least you should cite a work where the same protocol was used).

Discussion

4.       What is meant by metal ions characteristics (line 292)?

5.       Lines 293-296 should be rephrased for clarity.

Supplementary Tables TS1 ad TS2 are not available for review.

Minor misspellings and typos:

Line 23 – dyslipidemie – dyslipidaemia

Line 80-81 – Change for clarity to “Histological studies have shown the development of atherosclerotic-like degenerative changes in valves of patients with aortic valve stenosis”

Line 86 – did you mean ‘matrix’ instead of ‘mass’?

Line 88 – ‘mechanism’ should be ‘mechanisms’

Line 101 – space missing

Table 1 – replace the comma with a full stop on 0.776 (line “-Cannot be determined”)

Table 1 – remove double end brackets on AVA and AVAi

Line 158 – did you mean ‘demineralisation’?

Line 251 – remove the duplication ‘Hg and Cd bind’

Line 285 – delete duplicated full stop

Lines 287, 404 – full stop should be a comma

Line 322 – ‘predilection in the bones’ should be ‘predilection for bones’

Line 361 – full stop missing

Line 363- ‘heve’ should be ‘have’

Line 364 – ‘We did not use any such technique’ should be ‘We did not use any of such techniques’

Line 394 – ‘For he elements’ – t is missing

Line 411 – ‘analyzes’ should be ‘analyses’

Author Response

The study design and rationale are well explained, but I missed some details. Particularly:

  1. The variable ‘Occupational exposure’ should be better explained: what kind of jobs/activities were considered as an exposure risk? Add this to the manuscript.

Response: Occupational exposure for our patients were foundry, metalworking, welding, soldering and metalsmithing.

  1. The echocardiographic methods for assessing aortic stenosis severity and ejection fraction calculation should be indicated (for instance, did you use 2D or 3D echo? Did you use tissue Doppler? Did you use the modified Bernoulli equation to obtain the transvalvular gradients? Did you use Simpson’s biplane method for EF quantification? Did you follow the EAE guidelines?).

Thank you very much for your question.

Response: The structural changes of the affected valves and the severity of aortic valve stenosis were detected echocardiographically using Philips Ultrasound (machine) ie33 (Bothell, WA, USA) (18, 19, add new ref). Left ventricular ejection fraction (LVEF) was calculated by (using) the Simpson´s biplane method (2D echocardiography), transvalvular gradients (maximum and mean) and aortic valve area (AVA) were determined using continuous and pulsed Doppler parameters and instrument software (add new ref).

For the sake of simplicity, we have included only these parameters in Table 1. We have used the simplified Bernoulli equation in the past. Currently, most colleagues use instrument software to quantify gradients based on the basic Bernoulli equation with the determination of velocity time integrals (VTI). This procedure seems more accurate to us.

add new ref.: https://www.asecho.org/wp-content/uploads/2020/08/2009_Echo-Assessment-of-Valve-Stenosis_note_added.pdf

  1. The ICP-MS method used should be described in more detail (or at least you should cite a work where the same protocol was used).

Following text was added to the manuscript (line 164):

Median recoveries from replicate analysis of spiked samples and CRMs were found within the range of 93.9 and 110 % for spiked samples and 74.9 and 106 % for certified reference materials. Typical relative standard deviations within the replicate analysis of spiked samples were found between 1.8 and 10.4 %. Data were recorded by single analysis due to limited amount of sample available for analysis. Limits of detection (LODs) are dependent on amount of sample available for analysis (usually between 0.2 – 0.5 g). Typical values of LODs for elements in µg/g are as follows: 0.4 for Al, 0.0005 for Au, 0.01 for Ba, 0.0001 for Cd, 0.0003 for Co, 0.001 for Cr, 0.05 for Cu, 0.005 for Hg, 0.001 for Mo, 0.009 for Ni, 0.002 for Pb, 0.03 for Se, 0.009 for Sn, 0.008 for Sr, 0.06 for Ti, 0.0005 for V and 0.5 for Zn. LODs for elements in mg/g are as follows: 0.0001 for Mg, 0.003 for P, 0.09 for S and 0.002 for Ca.

Discussion

  1. What is meant by metal ions characteristics (line 292 PDF!)?

Response: The affinity of Cd, Hg and Zn ions to proline was monitored in vitro. In the aqueous phase, this metal ion characteristics decreased in the order Zn2+ > Cd2+ > Hg2+.

Corrected to:

In vitro, the affinity of Cd, Hg and Zn ions to proline was monitored, which in the aqueous phase decreases in the order Zn2+ > Cd2+ > Hg2+.

  1. Lines 293-296 should be rephrased for clarity.

In another experiment, even 24-hour exposure to Rh, Au, Co and Ni salts led to a reduction in collagen synthesis. Salt concentrations found in tissues surrounded by metallic materials were used. The effect of the other ions did not appear within 24 hours (35).

Rephrased to:

In another experiment, 24-hour exposure to Rh, Au, Co, and Ni salts at concentrations corresponding to those found in the tissues in the contact with metal material resulted in the reduction in collagen synthesis. The effect of other ions was not manifested in the first 24 hours (35).

----------------------------------------

Supplementary Tables TS1 ad TS2 are not available for review.

I do apologize. These ones have been sent as supplementary material. Uploaded once again directly.

Minor misspellings and typos:

Corrected in the manuscript. Thank you.

Line 23 – dyslipidemie – dyslipidaemia

Line 80-81 – Change for clarity to “Histological studies have shown the development of atherosclerotic-like degenerative changes in valves of patients with aortic valve stenosis”

Line 86 – did you mean ‘matrix’ instead of ‘mass’?

Line 88 – ‘mechanism’ should be ‘mechanisms’

Line 101 – space missing

Table 1 – replace the comma with a full stop on 0.776 (line “-Cannot be determined”)

Table 1 – remove double end brackets on AVA and AVAi

Line 158 – did you mean ‘demineralisation’?

Line 251 – remove the duplication ‘Hg and Cd bind’

Line 285 – delete duplicated full stop

Lines 287, 404 – full stop should be a comma

Line 322 – ‘predilection in the bones’ should be ‘predilection for bones’

Line 361 – full stop missing

Line 363- ‘heve’ should be ‘have’

Line 364 – ‘We did not use any such technique’ should be ‘We did not use any of such techniques’

Line 394 – ‘For he elements’ – t is missing

Line 411 – ‘analyzes’ should be ‘analyses’

Table TS1. Spearman-rank correlation between metals and trace elements in calcified aortic valve samples

Mg

P

Ca

S

Cr

Co

Cu

Zn

Se

Sr

Cd

Ba

Pb

Mo

Sn

Mg

0.57

0.58

-0.56

-0.02

-0.14

-0.48

0.48

-0.63

0.46

-0.12

0.33

0.24

-0.09

0.21

P

0.57

0.99

-0.77

-0.24

-0.49

-0.55

0.41

-0.49

0.54

-0.51

0.23

0.09

-0.10

0.01

Ca

0.58

0.99

-0.80

-0.24

-0.50

-0.60

0.43

-0.52

0.54

-0.50

0.23

0.10

-0.10

-0.01

S

-0.56

-0.77

-0.80

0.26

0.39

0.60

-0.41

0.76

-0.42

0.39

-0.12

-0.21

-0.08

0.04

Cr

-0.02

-0.24

-0.24

0.26

0.37

0.10

-0.05

0.02

0.12

0.02

0.10

0.17

0.07

0.26

Co

-0.14

-0.49

-0.50

0.39

0.37

0.41

-0.35

0.27

0.02

0.36

0.06

0.07

0.06

0.08

Cu

-0.48

-0.55

-0.60

0.60

0.10

0.41

-0.49

0.60

-0.29

0.24

-0.15

-0.17

0.24

-0.04

Zn

0.48

0.41

0.43

-0.41

-0.05

-0.35

-0.49

-0.33

0.35

-0.23

0.31

0.37

0.04

0.14

Se

-0.63

-0.49

-0.52

0.76

0.02

0.27

0.60

-0.33

-0.30

0.09

-0.05

-0.29

-0.04

0.12

Sr

0.46

0.54

0.54

-0.42

0.12

0.02

-0.29

0.35

-0.30

-0.20

0.44

0.18

-0.07

0.25

Cd

-0.12

-0.51

-0.50

0.39

0.02

0.36

0.24

-0.23

0.09

-0.20

-0.03

0.04

-0.04

-0.20

Ba

0.33

0.23

0.23

-0.12

0.10

0.06

-0.15

0.31

-0.05

0.44

-0.03

0.33

-0.22

0.00

Pb

0.24

0.09

0.10

-0.21

0.17

0.07

-0.17

0.37

-0.29

0.18

0.04

0.33

0.19

-0.32

Mo

-0.09

-0.10

-0.10

-0.08

0.07

0.06

0.24

0.04

-0.04

-0.07

-0.04

-0.22

0.19

-0.04

Sn

0.21

0.01

-0.01

0.04

0.26

0.08

-0.04

0.14

0.12

0.25

-0.20

0.00

-0.32

-0.04

Table TS2. Spearman-rank correlation between metals and trace elements in aortic valve samples from controls

Mg

P

Ca

S

Cr

Co

Cu

Zn

Se

Sr

Cd

Ba

Pb

Mo

Sn

Mg

0.94

0.71

0.54

0.25

0.43

0.47

0.79

0.25

0.63

0.39

0.61

0.69

0.48

0.15

P

0.94

0.84

0.37

0.23

0.43

0.32

0.68

0.08

0.74

0.51

0.68

0.77

0.39

0.17

Ca

0.71

0.84

-0.01

0.19

0.38

-0.04

0.44

-0.27

0.90

0.41

0.73

0.78

0.08

0.14

S

0.54

0.37

-0.01

0.01

0.17

0.81

0.62

0.77

-0.11

0.09

0.01

0.12

0.44

0.19

Cr

0.25

0.23

0.19

0.01

0.33

-0.02

0.32

0.17

0.26

0.23

0.28

0.20

0.06

0.00

Co

0.43

0.43

0.38

0.17

0.33

0.15

0.29

0.10

0.45

0.22

0.53

0.48

0.15

0.33

Cu

0.47

0.32

-0.04

0.81

-0.02

0.15

0.63

0.63

-0.15

0.21

0.00

0.01

0.64

0.10

Zn

0.79

0.68

0.44

0.62

0.32

0.29

0.63

0.50

0.34

0.37

0.36

0.49

0.61

0.20

Se

0.25

0.08

-0.27

0.77

0.17

0.10

0.63

0.50

-0.26

0.16

-0.17

-0.10

0.52

0.13

Sr

0.63

0.74

0.90

-0.11

0.26

0.45

-0.15

0.34

-0.26

0.33

0.82

0.77

0.05

0.00

Cd

0.39

0.51

0.41

0.09

0.23

0.22

0.21

0.37

0.16

0.33

0.25

0.48

0.45

-0.11

Ba

0.61

0.68

0.73

0.01

0.28

0.53

0.00

0.36

-0.17

0.82

0.25

0.69

0.02

0.16

Pb

0.69

0.77

0.78

0.12

0.20

0.48

0.01

0.49

-0.10

0.77

0.48

0.69

0.12

0.14

Mo

0.48

0.39

0.08

0.44

0.06

0.15

0.64

0.61

0.52

0.05

0.45

0.02

0.12

-0.13

Sn

0.15

0.17

0.14

0.19

0.00

0.33

0.10

0.20

0.13

0.00

-0.11

0.16

0.14

-0.13

Round 2

Reviewer 1 Report

I am happy to see the manuscript improved nicely. The authors addressed all my comments adequately.
However, to make sure the statements are supported by literature I will recommend adding citations in the following lines, and I recommend it for publication after the minor comments.:

Line 89-92:  https://doi.org/10.1038/nrdp.2016.6

Line 239-246:
https://doi.org/10.1007/s10661-020-08818-w

Line 221-224: https://doi.org/10.3390/ijerph7072745
or
https://doi.org/10.1016/j.ccr.2020.213343